# Topography influences diurnal and seasonal microclimate fluctuations in hilly terrain environments of coastal California

**Aji John**[1]*, **Julian D. Olden**[2], **Meagan F. Oldfather**[3], **Matthew M. Kling**[4], **David D. Ackerly**[4,5]

1 Department of Biology, University of Washington, Seattle, WA, United States of America, 2 School of Aquatic and Fishery Sciences, University of Washington, Seattle, WA, United States of America, 3 Department of Ecology and Evolutionary Biology, University of Colorado, Boulder, CO, United States of America, 4 Department of Integrative Biology, University of California–Berkeley, Berkeley, CA, United States of America, 5 Department of Environmental Science, Policy and Management, University of California–Berkeley, Berkeley, CA, United States of America

* ajijohn@uw.edu

**Data Availability Statement:** The data underlying the results presented in the study are available on

## Abstract

Understanding the topographic basis for microclimatic variation remains fundamental to predicting the site level effects of warming air temperatures. Quantifying diurnal fluctuation and seasonal extremes in relation to topography offers insight into the potential relationship between site level conditions and changes in regional climate. The present study investigated an annual understory temperature regime for 50 sites distributed across a topographically diverse area (>12 km$^2$) comprised of mixed evergreen-deciduous woodland vegetation typical of California coastal ranges. We investigated the effect of topography and tree cover on site-to-site variation in near-surface temperatures using a combination of multiple linear regression and multivariate techniques. Sites in topographically depressed areas (e.g., valley bottoms) exhibited larger seasonal and diurnal variation. Elevation (at 10 m resolution) was found to be the primary driver of daily and seasonal variations, in addition to hillslope position, canopy cover and northness. The elevation effect on seasonal mean temperatures was inverted, reflecting large-scale cold-air pooling in the study region, with elevated minimum and mean temperature at higher elevations. Additionally, several of our sites showed considerable buffering (dampened diurnal and seasonal temperature fluctuations) compared to average regional conditions measured at an on-site weather station. Results from this study help inform efforts to extrapolate temperature records across large landscapes and have the potential to improve our ecological understanding of fine-scale seasonal climate variation in coastal range environments.

## Introduction

Human-induced climate change is a major cause of species extinctions and biodiversity loss globally [1]. Widespread species range shifts in response to warming are already evident [2,3],

Zenodo (Link: https://doi.org/10.5281/zenodo.
10806625).

**Funding:** The author(s) received no specific funding for this work.

**Competing interests:** The authors have declared that no competing interests exist.

and substantial additional reshuffling of plant and animal communities is likely [4]. However, mounting evidence suggests that small scale topographic variation promotes high levels of climatic heterogeneity [5,6], potentially mediating expected large scale range shifts caused by macroclimatic change [2,7,8]. Facilitated by unique micro-topographic characteristics [8,9], this can offer opportunities for species to persist in place, rather than shift in space, in response to climate change [10–13].

Topography and vegetation create microclimates that differ from regional climates. These local departures can be explained by physio-topographic attributes such as elevation, slope and aspect, depressions and ridgetops, canopy cover, distance to coast, and proximity to the forest edge [14–16]. These features give rise to spatio-temporal heterogeneity expressed on local scales, affecting minimum and maximum temperatures, and the resulting magnitude of diurnal (daily maximum vs. minimum) and seasonal (summer vs. winter) temperature variations [9,17]. Furthermore, the resulting minimum and maximum extreme temperatures, are known to structure plant and animal distributions [18–20]. In topographically heterogeneous landscapes in particular, steep topographic gradients create strong microclimate heterogeneity [21].

One effect that may occur locally is microclimatic buffering at diurnal, seasonal or interannual scales. We define buffering as reduced temperature fluctuations relative to free atmospheric conditions measured by e.g., weather stations [11,22]. Buffering capacity, in the context of our study, refers to the ability of a specific location to moderate or lessen temperature fluctuations [6,23]. When a location has a high buffering capacity, it can effectively reduce the magnitude of temperature variations, providing more stable and moderated temperature conditions. Alternatively, a reduced buffering capacity implies that the location is less effective at mitigating temperature fluctuations, resulting in more pronounced and variable temperature changes [20]. Vegetation canopies have important buffering effects, with lower daytime maxima and higher nighttime minima [24–26], and buffering of temperature extremes in comparison with a forest clearing e.g., clearcuts [27–29].

Cold-air pooling modifies the coupling of sites with free-air temperature in conjunction with physiographic features. Coupling is defined as having a 1:1 linear relationship with the free-air temperature (e.g., clearcut site or a weather station) [22]. The cold-air pooling conditions keep cold air trapped in convergent environments like valley bottoms, yielding stable microclimates in some circumstances [30]. Generally, the relationship between temperature and elevation (or altitude) is explained by the adiabatic lapse rate of 6–8°C decrease per 1000 meters increase in elevation. However, ridgetops and valley bottoms can influence temperatures via orographic effects, causing adiabatic cooling on windward slopes and potential warming on leeward sides [31,32]. By contrast, the higher density of cold air can lead to accumulation in low-lying regions, and to temperature inversions with warm air resting above layers of cold air [33]. This is predominantly manifested during the winter on clear nights with light winds. Hence, cold-air pooling is a key determinant of the degree of coupling between the boundary layer and free atmosphere [32,34].

Previous research has pointed to the role of micro-topographic and vegetative features in buffering or coupling of microclimates with respect to regional climate. However, what remains less clear is the way different landscape patterns (characterized by topographic features and vegetation) influence mean diurnal and seasonal temperature variations as such. In fact, the potential impacts of temperature variability and extremes can pose a greater risk to species than increases in mean temperature [35,36]. It is likely that organisms occupying sites that have higher seasonal and diurnal fluctuations may exhibit greater tolerance of heat extremes and greater potential to withstand climate change impacts. This notion is not new: Janzen (1967) predicted decades ago that species experiencing large environmental variability

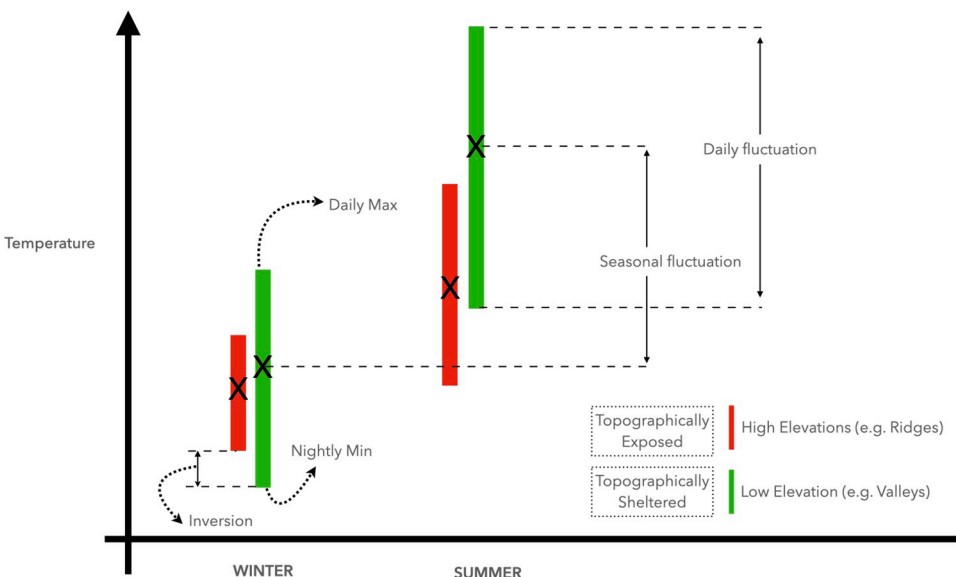

**Fig 1. Conceptual diagram illustrating possible topographic effects on diurnal and seasonal fluctuations.**

will acclimate better and have increased range limits than the ones in lower variability regimes. Greater temperature fluctuations may better prepare species to tolerate future warming as they have evolved or acclimated to broader climatic tolerances.

Here, we study a large mixed-hardwood forest landscape in Northern California to determine the role of topography and vegetation in diurnal and seasonal fluctuations, with a focus on canopy cover, slope, aspect, and hillslope position. We specifically examine how topography drives diurnal and seasonal minimum, maximum, and mean temperatures, and how it influences resulting diurnal and seasonal temperature fluctuations. We hypothesize that valley bottoms will show greater temperature fluctuations, due to the combination of cold-air pooling and reduced diurnal mixing with the local free atmosphere [32,37,38], leading to reduced temperature buffering capacity compared to ridge-tops (conceptual model in Fig 1). Knowing how species respond to changes in diurnal temperature fluctuations is crucial for understanding their adaptation to climate [39,40].

## Methods

### Study design

The study was conducted at Pepperwood Preserve (Sonoma Co., 38.57°N, 122.68°W) in northern California (Fig 2: Pepperwood Preserve Study Area, Fig B.4 in S1 Appendix with site numbers). The preserve is representative of deciduous and evergreen woodlands in the region, and exhibits elevations from 122 m to 462 m, with rugged undulating topography. The local climate is coastal Mediterranean, with hot summers, cool winters, and predominantly winter precipitation. The long-term average temperature at the study site was found to be 15.37°C, with a standard deviation of 0.53°C. In contrast, the regional temperature is found to be 14.49°C with a standard deviation of 0.55°C. These values signify the variability in temperature over the past 40 years.

### Climate data collection

Fifty 20 x 20 m vegetation plots were situated by stratifying on micro-topographic features and vegetation types [41]. In the summer of 2013, 50 temperature data loggers (Onset HOBO Model U23, Onset Corp., Bourne, MA) were installed at each plot. Each logger was placed 1.2

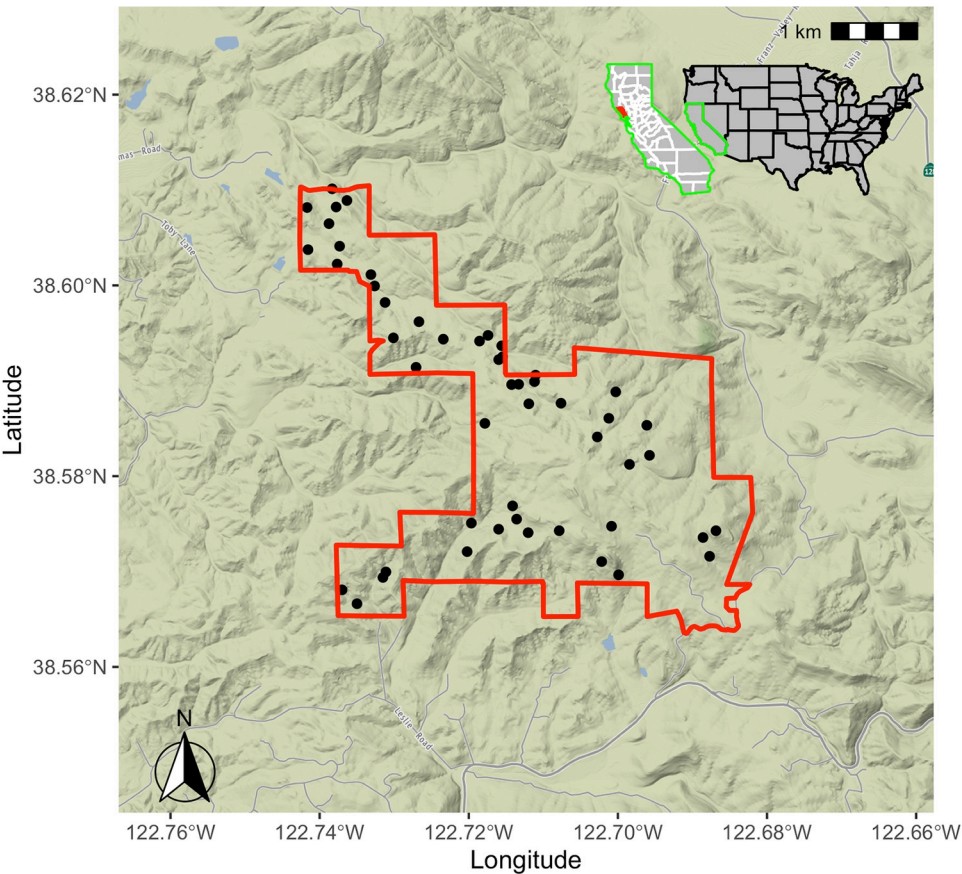

**Fig 2. Pepperwood Preserve with 50 study sites.** Each site was equipped with a microclimate logger (Onset HOBO Model U23, Onset Corp., Bourne, MA) that was installed at 1.2m height above the ground (see Fig A.3 in S1 Appendix for a picture of the logger). Image credits © Stamen Design, © OpenStreetMap.

m above the ground inside a radiation shield (Fig A.3 in S1 Appendix). Loggers were placed at the edge of each plot, and under representative canopy cover conditions. All of the loggers were placed in the understory and are part of a long-term forest dynamics research project [41,42]. Data loggers recorded temperature (˚C) and relative humidity (RH%) at half-hour intervals. At each site, average hourly temperatures were first calculated at the site level, and then from those, average daily temperatures (minimum, maximum and diurnal) were calculated for winter (December 2013-February 2014), spring (March-May 2014), summer (June-August 2014) and autumn (Sep-Nov 2014) months. From these daily metrics, 18 summary statistics were calculated (Table 1) representing daily minimum, mean, maximum, and diurnal variation in each season, plus growing degree day accumulation and the variation in seasonal means (difference between summer mean and winter mean). A permanent weather station at the Pepperwood Preserve (situated in an open area) was used as a reference in our study. To evaluate coupling and buffering of a site with regional climate (in this case. we used a weather station), quantification was done following methods of De Frenne et al. 2021.

## Topographic and canopy characterization

Spatial analysis using a GIS [43] was performed on a 10 m digital elevation model (DEM) of Pepperwood Preserve [41] to derive topographic variables (for details on topographic variables

**Table 1. Derived climate variables and their minimum, mean, and maximum across the 50 logger sites.**

| Description | unit | abbreviation | min, max (mean) |
|---|---|---|---|
| Mean summer daily diurnal variation (Max–Min) | ˚C | mean_fluct_summer | 11.56, 19.75 (15.96) |
| Mean daily min temperature in summer. | ˚C | mean_summer_min | 11.30, 13.93 (12.28) |
| Mean daily max temperature in summer. | ˚C | mean_summer_max | 25.48, 31.60 (28.23) |
| Mean daily temperature in summer. | ˚C | summer_mean | 18.04, 20.38 (19.06) |
| Mean winter daily diurnal variation. | ˚C | mean_fluct_winter | 4.44, 15.78 (9.57) |
| Mean daily min temperature in winter. | ˚C | mean_winter_min | 1.94, 9.56 (6.64) |
| Mean daily max temperature in winter. | ˚C | mean_winter_max | 12.78, 23.61 (16.21) |
| Mean daily temperature in winter. | ˚C | winter_mean | 8.19, 14.68 (10.75) |
| Growing degree days accumulated in the growing season. Defined as thermal units accumulated above 10˚C | degree-days | gdd | 1616, 2675 (2191) |
| Mean spring daily diurnal variation. | ˚C | mean_fluct_spring | 8.64, 17.64 (12.59) |
| Mean daily min temperature in spring. | ˚C | mean_spring_min | 6.60, 10.63 (8.74) |
| Mean daily max temperature in spring. | ˚C | mean_spring_max | 17.06, 27.37 (21.33) |
| Mean daily temperature in spring. | ˚C | spring_mean | 11.53, 17.68 (14.28) |
| Mean autumn daily diurnal variation. | ˚C | mean_fluct_autumn | 7.85, 16.30 (12.01) |
| Mean daily min temperature in autumn. | ˚C | mean_autumn_min | 8.79, 13.37 (11.11) |
| Mean daily max temperature in autumn. | ˚C | mean_autumn_max | 20.90, 26.40 (23.13) |
| Mean daily temperature in autumn. | ˚C | autumn_mean | 14.28, 17.12 (16.05) |
| Seasonal mean fluctuation (summer mean–winter mean) | ˚C | seasonal_mean_fluct | 4.77, 11.31 (8.31) |

used in this study see Oldfather et al. 2016) (Table 2 and Table A.2 in S1 Appendix). Variables included TPI (topographic position index), which is calculated as the difference (m) between the elevation of a point and the mean elevation in a surrounding radius. This measure indicates how elevated or depressed a site is in relation to its surroundings (positive for ridges and hilltops, and negative for valley bottoms). 100 m and 500 m radii were used to produce TPI100 and TPI500, which capture fine-scale variations and larger terrain features, respectively. TPI is essentially a local elevation, in contrast to DEM which is the global elevation. Similarly, PLP (percent lower points) is defined as the percentage of points within a given radius that are lower than the focal point (higher percentages indicate hilltops, and lower percentages identify valleys). PLP and TPI measures at each scale were strongly correlated (pairwise $r^2$ ranged from .84 to .94); furthermore, the metrics calculated using 500 m were similar to 100 m, so only PLP500 was used in this study [41]. Slope and aspect were calculated using the *raster* R package [44]. Northness was calculated as cosine(aspect)*sine(slope) [42] (aspect on its own was examined but did not contribute to the final analysis). Pairwise Pearson correlation coefficients among topographic variables ranged from -0.26 to 0.29, with the highest between DEM and PLP500, as these are two measures of site elevation. Canopy cover was recorded in mid-

**Table 2. Topographic and canopy variables measured at each of the 50 sites.**

| Description | unit | abbreviation | min, max (mean) |
|---|---|---|---|
| Elevation derived from Digital Elevation Model (referred as elevation in the text) | m | DEM (Elevation) | 121.89, 461.52 (272.02) |
| Percent of pixels lower in a 500m radius | % | PLP500 (Topographic position) | 6.14, 99.85 (47.15) |
| Measure of steepness | ° | slope | 3.55, 29.77 (18.80) |
| Canopy cover | % | canopy | 32.85, 91.03 (74.72) |
| Northness (cos(aspect)*sin(slope)) | | northness | -0.43, 0.42 (0.06) |

summer when the plots were initially established, using a forest densiometer [41]. Note that plots have varying proportions of deciduous trees (0 to 100%); the cover values are thus most relevant for analysis of summer temperatures. Canopy cover for winter was modified by subtracting the deciduous fraction from the total canopy cover.

### Statistical analyses

Each site is a 20 x 20 m plot and constitutes a study unit. Two data matrices were produced for analysis; the climate matrix contained all the climatic descriptors for the sites (50 sites x 18 variables), and the physiographic matrix included topographic and canopy related variables (50 sites x 5 variables). All the variables were continuous. Two sites (Site 1348 and 1350) were missing climate data for roughly six months but were still included in all the analyses for completeness, and removing or including them did not alter the outcome.

To reduce the dimensionality and to assess dominant variables in climate and physiographic space, principal component analysis (PCA) was employed. PCA is known to work better than stepwise regression in multivariate datasets where high correlations exist between the variables. Correlation matrix was used for PCA as variables differed in their measurement units. PCA was performed separately on the physiographic and climate matrices. A Monte Carlo randomization test was used to assess the statistical significance of each principal component (PC) axis.

Multiple linear regression was performed to tease apart variations in climatic space for individual seasons. Additionally, principal component regression (PCR) was used to assess the association between physiographic characteristics (summarized according to PC1 and PC2) and the minimum, maximum, mean, and diurnal variations in summer and winter temperatures. A complementary redundancy analysis (RDA) was also conducted to investigate the relationship between the entire suite of climate and physiographic variables. We further performed selective RDA by subsetting seasonal (winter, summer, autumn and spring) climate variables and seeking out the dominant topographic variables contributing to corresponding diurnal variations in different seasons. Aggregate climate metrics (min, max, mean, and seasonal fluctuations) were normalized for RDA analysis using log-10 transformation.

All analyses were performed using R software [45] using several R packages, including '*tidyverse*', '*raster*' and '*vegan*' [44,46,47].

## Results

Over the 12-month cycle (December 2013-November 2014), temperatures across the landscape varied from a minimum of -2.7˚C to a maximum of 40.4˚C; mean hourly temperatures across the seasons (over all the days in a season) were between 9˚C and 30˚C (Fig 3A–3C), with an overall site mean temperature of 15˚C (over all sites and days). Across the landscape, altitude was positively correlated with average temperature (Fig B.3 in S1 Appendix; $R^2$ = 0.23, p < 0.001). Sites across the Pepperwood Preserve show significant diurnal variations in different seasons. In summer, the average diurnal variation was approximately 16˚C and in winter it was approximately 9˚C. Winter maximum temperatures exhibited the greatest spatial variability across sites (Fig 3D).

Multiple regression analysis revealed that seasonal climate metrics were strongly related to elevation and moderated by micro-topographic and canopy features (Table 3). Diurnal variations in each season decreased with DEM (elevation), indicating that higher elevations exhibited lower diurnal variations than did lower elevation sites. Diurnal variations for autumn, spring and summer (all seasons except winter) were correlated negatively with PLP500 and canopy cover. Diurnal variation was also found to be negatively associated with northness in

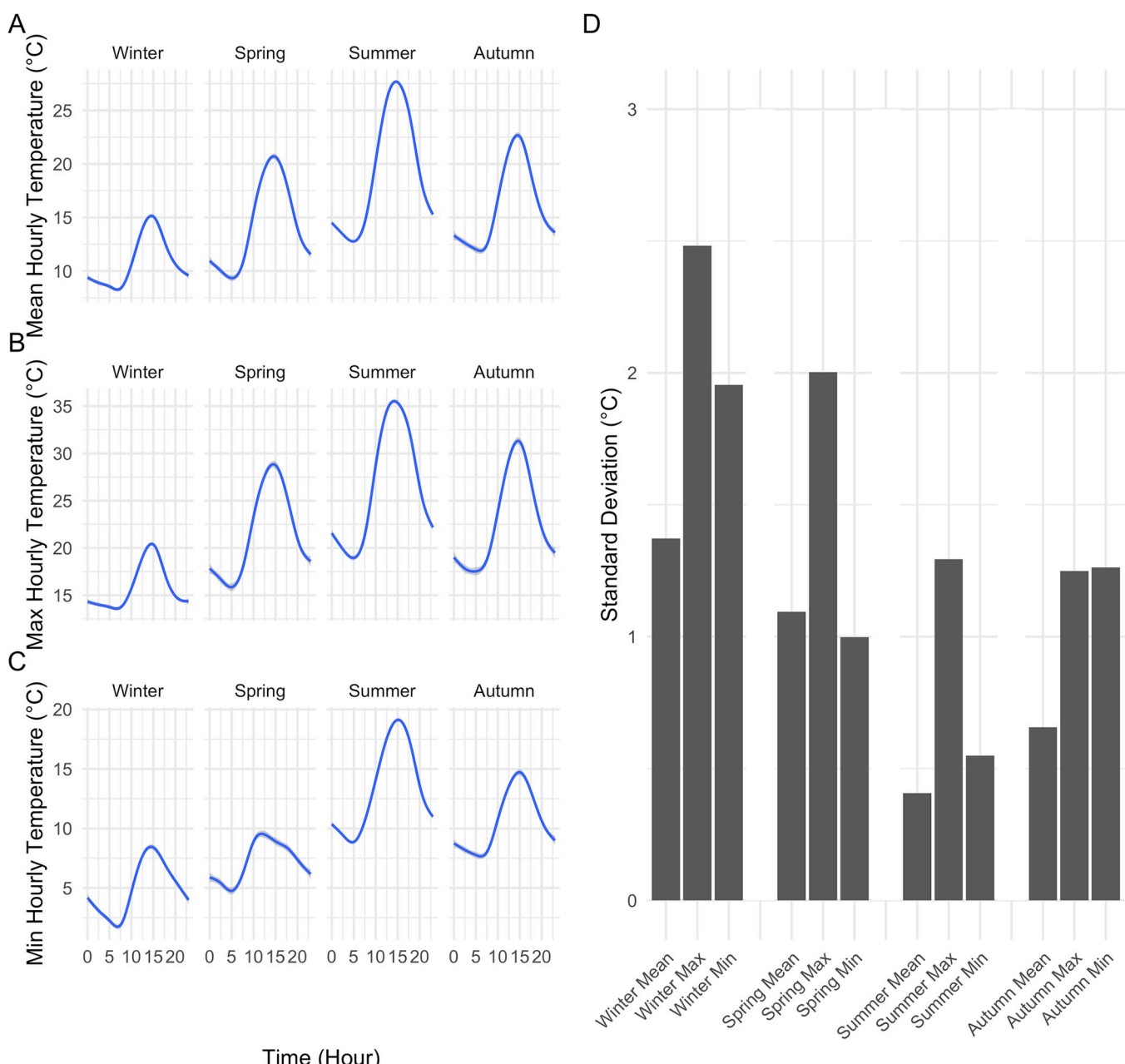

**Fig 3. Seasonal minimum, maximum average temperatures with corresponding variability (lines are loess fits).** (A) Mean hourly temperatures of 50 sites by season. (B) Maximum hourly temperatures of 50 sites by season. (C) Minimum hourly temperatures of 50 sites by season. (D) Standard deviation of seasonal mean, minimum and maximum across sites.

autumn only. Seasonal minimum temperatures were positively associated with elevation, suggesting temperature inversions, in contrast with max temperatures that were negatively associated with elevation. Minimum temperatures were positively associated with global elevation (DEM) and local elevation or hillslope position (PLP500), except in winter where it was DEM only. Seasonal maximum temperatures were negatively associated with elevation but were found to be moderated by canopy cover in summer and in autumn. Additionally, autumn and winter maximum temperatures were negatively associated with northness. Seasonal

**Table 3. Partial coefficients from the multiple linear regression of climate variables by topographic drivers (*p-value* significance codes in bold: '\*\*\*' < 0.001, '\*\*' < 0.01, '\*' < 0.05, '.' < 0.1, " < 1).**

| Variable | Beta coefficients of main physiographic variables | | | | | R-Squared |
|---|---|---|---|---|---|---|
| | DEM | Slope | Canopy | PLP500 | Northness | |
| mean_fluct_winter | **-0.019\*\*\*** | 0.047 | -0.018 | -0.016 | -3.284 | 0.42 |
| mean_fluct_spring | **-0.015\*\*\*** | -0.025 | **-0.036\*** | **-0.026\*\*** | -0.451 | 0.68 |
| mean_fluct_summer | **-0.011\*\*\*** | 0.003 | **-0.050\*\*\*** | **-0.014\*** | 0.266 | 0.72 |
| mean_fluct_autumn | **-0.016 \*\*\*** | -0.006 | **-0.041\*\*\*** | **-0.022\*\*\*** | **-1.781\*\*** | 0.86 |
| winter_mean | **0.007\*\*** | 0.041 | -0.006 | 0.009 | -1.292 | 0.34 |
| spring_mean | -0.003 | -0.022 | -0.012 | 0.010 | -0.013 | 0.11 |
| summer_mean | -0.000 | -0.008 | **-0.021\*\*\*** | **0.005\*** | 0.098 | 0.59 |
| autumn_mean | **0.003\*\*\*** | -0.004 | **-0.014\*\*** | **0.014\*\*\*** | **-1.041\*\*\*** | 0.64 |
| winter_min | **0.016\*\*\*** | 0.026 | 0.000 | 0.014 | -0.504 | 0.62 |
| spring_min | **0.004\*\*** | -0.011 | 0.002 | **0.022\*\*\*** | -0.288 | 0.51 |
| summer_min | **0.003\*\*\*** | -0.005 | 0.002 | **0.013\*\*\*** | -0.202 | 0.66 |
| autumn_min | **0.009\*\*\*** | -0.003 | 0.002 | **0.023\*\*\*** | -0.717 | 0.80 |
| winter_max | -0.003 | 0.073 | -0.018 | -0.002 | **-3.788\*** | 0.21 |
| spring_max | **-0.011\*\*\*** | -0.036 | -0.034 | -0.004 | -0.739 | 0.36 |
| summer_max | **-0.008\*\*\*** | -0.002 | **-0.048\*\*\*** | -0.001 | 0.065 | 0.66 |
| autumn_max | **-0.008\*\*\*** | -0.010 | **-0.040\*\*\*** | 0.001 | **-2.498 \*\*\*** | 0.67 |
| seasonal_amplitude (summer and winter) | **-0.007\*\*\*** | -0.051 | -0.016 | -0.002 | 1.144 | 0.38 |
| gdd | -0.224 | -4.976 | **-8.044\*\*\*** | **3.328\*\*** | **-375.866\*\*** | 0.44 |

fluctuations (summer mean minus winter mean) decreased with elevation only. Slope was not significantly related to any of the variables.

Significant proportions of the variability were captured by the first two PCs in climate and physiographic space. In the first PCA (climate), PC1 and PC2 explained 37.2% and 30.0%, respectively, of the climatic variability across Pepperwood Preserve (Fig 4A). Variables with positive loadings on PC1 included mean minimum temperatures from winter, summer, and autumn, and winter and autumn mean temperatures. Variables with negative loadings on PC1 included summer and winter fluctuations and seasonal maxima. Negative loadings on PC2 included GDD, and several seasonal means and maxima. In the ensuing PCA results, we show the $R^2$ of the respective correlations in parentheses.

In the second PCA (physiography), PC1 and PC2 explained 54.3% (27.2% and 27.1% respectively) of the topographic variability across Pepperwood Preserve (Fig 4B). Topography-related variables were positively correlated with PC1 except for DEM which was negatively correlated: DEM (R = -0.77), slope (R = 0.42), northness (R = 0.42) and PLP 500 (R = 0.07). Canopy was positively correlated on PC1 (R = 0.74). Hence, positive values of PC1 represent sites that are low elevation north facing ridges with steep slopes and denser canopy. PC2 of physiographic space was largely composed of positively correlated topographic variables: DEM (R = 0.35), PLP500 (R = 0.88) and northness (R = 0.63). Canopy and slope were negatively correlated on PC2 (R = -0.20 & -0.14). Positive values of PC2 can be interpreted as low canopy cover sites on higher elevations (ridgetops).

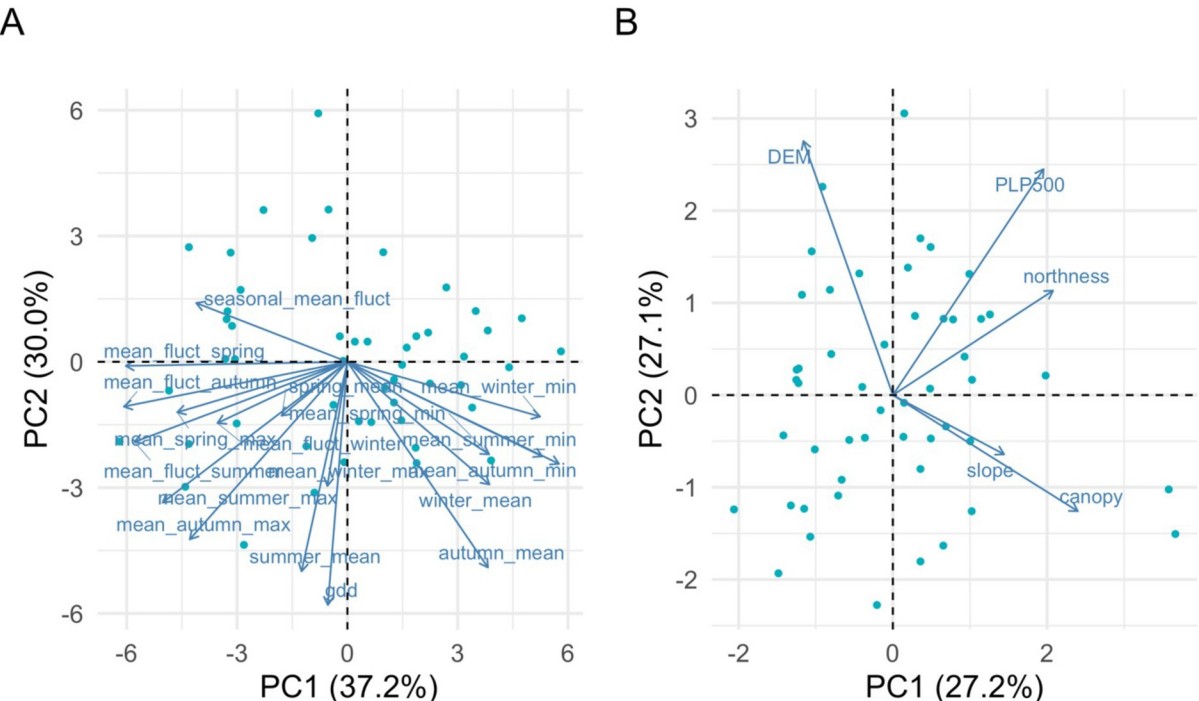

**Fig 4.** Biplots of climate and physiographic PCA analysis (A) PCA of climate space (B) PCA of physiographic space. Dots on the panels represent sites, and arrows represent variables, with their length indicating their contribution to the principal component.

A multivariate linear regression model with the physiographic PCs was explored against all the climate variables (Figs A.1 and A.2 in S1 Appendix). Summer mean and summer maximum temperatures were positively correlated with PC1 of physiographic PCA axes ($R^2 = 0.37$ & $R^2 = 0.23$, p-value < 0.0001), summer diurnal variations were marginally correlated ($R^2 = 0.13$, p-value < 0.05), but winter temperature measures were not correlated with PC1. PC2 of topographic space reflected the gradient from valley bottoms to high elevation sites and explained variation in winter temperature measures. Winter minimum and winter mean temperatures were positively correlated with PC2 of physiographic PCA axes ($R^2 = 0.42$ & $R^2 = 0.09$, p-value < 0.0001 and p-value < 0.05), but winter diurnal variations and winter maximum temperatures were negatively correlated ($R^2 = 0.37$ & $R^2 = 0.07$, p-value < 0.0001 and p-value < 0.05) with PC2 of physiographic PCA axes (Fig A.2 in S1 Appendix).

RDA confirmed the strong influence of topographic variables on climate variability. RDA results show that topographic variables DEM, PLP500, northness and slope explained a significant proportion of climatic variability across all the seasons (Fig 5). Table A.1 in S1 Appendix shows the RDA output when all the topographic variables were selectively constrained against seasonal climate variables. In autumn, statistical significance (p-value < 0.005) is observed for topographic features such as DEM, PLP500, and northness, as well as canopy. However, in spring, only topographic features DEM and PLP500 demonstrate significance (p-value < 0.005). In summer, DEM, PLP500 and canopy are found to be statistically significant (p-value < 0.005), and in winter, only DEM is found to be statistically significant (p-value < 0.005). Additionally, temperature metrics (maximum and diurnal) overlap with canopy and slope explaining the seasonal variation in all seasons, but canopy plays an important role in summer (Fig 5). RDA analysis shows that DEM is the most dominant topographic variable which explains the variation in climate across a heterogeneous landscape.

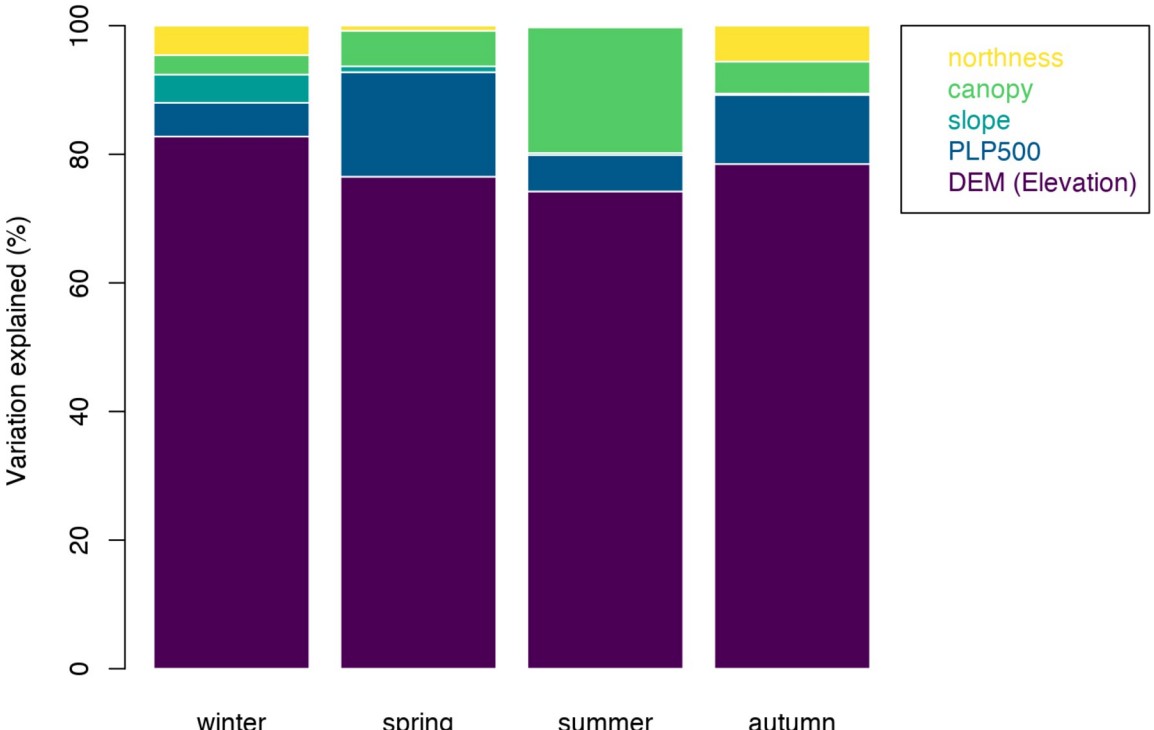

**Fig 5. Topographic variables contributions deduced by RDA across all the seasons.** Elevation was found to be a prominent driver of variability in all seasons, with smaller contributions from other topographic features.

Inversion of minimum temperatures was found across Pepperwood Preserve in all the seasons. Lower elevation sites are colder than the higher elevation sites, especially in winter as evidenced by the positive association of minimum temperatures with elevation (Fig 6A). Autumn and spring are intermediate, with a slightly stronger pattern in autumn (Fig 6A). Unlike the minima, maximum temperatures decline with elevation, and the pattern was weakest in winter (Fig 6B). Higher elevation sites have lower diurnal variation in all seasons as opposed to lower elevation valley sites (Fig 7A). Across all the seasons, diurnal variations were highest in summer with the lowest experienced in winter.

Comparison of daily mean temperatures of HOBOs with Pepperwood weather station showed a general trend of buffered temperatures in the understory compared to the weather station (Fig 8, Fig B.1 in S1 Appendix). A majority of sites were found to be cooler in spring and summer, but warmer in fall and winter. The pattern was not evident in some of the high elevation sites (cluster of sites around 400 m elevation) where it tended to be nearly aligned with the weather station (Fig 8B). Furthermore, comparison of diurnal variation of loggers with weather station shows lower diurnal variation of valley bottom sites than the high elevation sites with respect to weather station (Fig B.2 in S1 Appendix)

## Discussion

Our study supports the general understanding that elevation is the primary driver of temperature variation, but also illustrates the additional influence of local vegetation and topography on the climatic conditions manifested at the site level in mountainous terrain. We confirm the role of elevation at a larger landscape level influencing both diurnal and seasonal temperature fluctuations [9,48], but found it to be inverted for minimum temperatures and following a

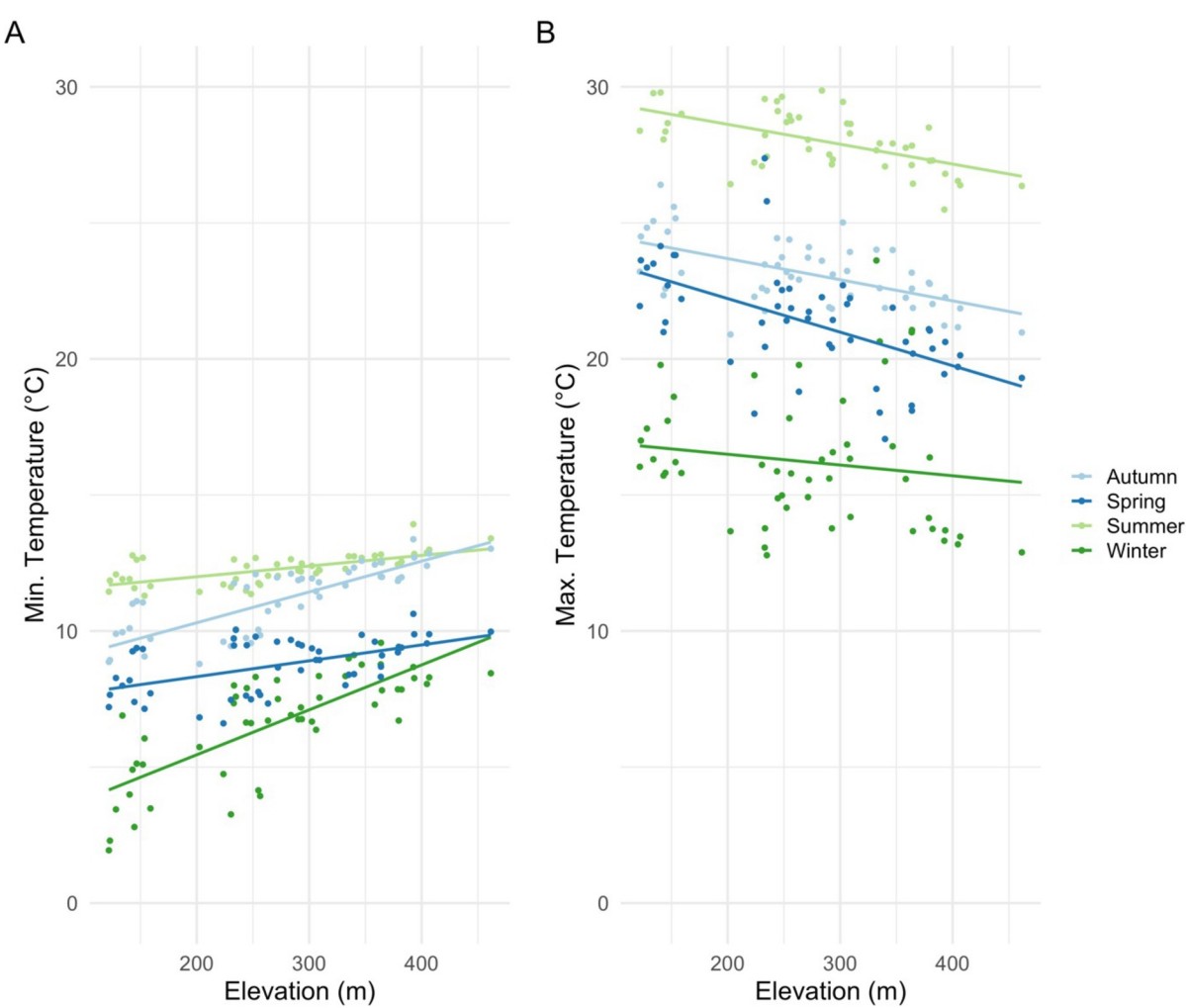

**Fig 6. Linear fits of seasonal minima and maxima across the elevational gradient.** (A) Lower elevation sites are found to be colder than the high elevation suggesting inversion (temperature increasing with elevation); autumn minimum temperatures exhibiting higher rate of increase by elevation than in the spring; autumn ($R^2 = 0.65$), spring ($R^2 = 0.28$), summer ($R^2 = 0.41$), and winter ($R^2 = 0.58$) all significant at p-value < 0.001; (B) Summer maximum temperatures are found to be declining with elevation, but is less pronounced in winter; maximum temperatures association with elevation in autumn is found to be less pronounced compared to spring; autumn ($R^2 = 0.31$), spring ($R^2 = 0.31$), summer ($R^2 = 0.45$) significant at p-value <0.001, and winter ($R^2 = 0.02$) not significant at p-value = 0.31.

gradient for maximum temperatures (Fig 6A and 6B). Additionally, topographic features like topographic position (PLP500, direct measure of hilltop/valley bottoms) and northness contributed to the diurnal and seasonal variability, minima, and maxima. We also revealed that canopy mediates the diurnal variation in different seasons [49]. Valley-bottom sites were in a cold-air pool and showed more variability (diurnal and seasonal), thus supporting our main hypothesis. Given that our results align with general understanding of elevation as the key factor in temperature, its interaction with more detailed physiographic features influence temperature variation [9,50]. This supports the validity of downscaling approaches that use coarse-grain climate grids to produce fine-grained products using only the elevation [51,52]. However, one needs to be cautious as seasonal and region-specific lapse rates like shown here are to be considered when downscaling [31]. Additionally, our study only used a year of climate data to look at topographic effects; a longer time period is warranted to confirm the uncovered patterns.

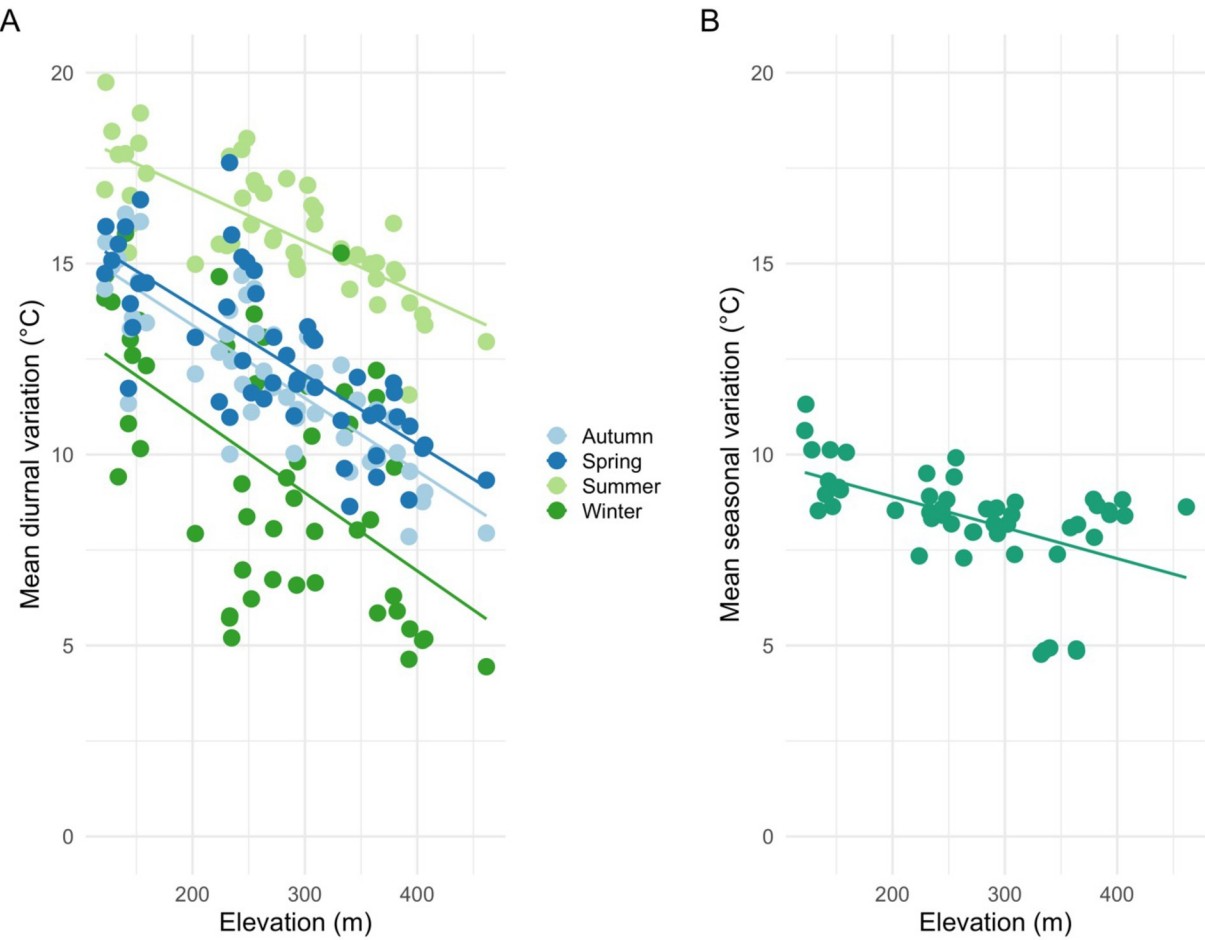

**Fig 7. Linear fits of seasonal diurnal fluctuations as a function of elevation.** Higher elevation sites exhibit lower diurnal variation in all seasons; autumn ($R^2 = 0.69$, p-value < 0.001), spring ($R^2 = 0.57$, p-value < 0.001), summer ($R^2 = 0.57$, p-value < 0.001), and winter ($R^2 = 0.31$, p-value < 0.001) (A) and lower seasonal variation (difference between mean summer and winter fluctuation); $R^2 = 0.25$, p-value < 0.001(B).

Diurnal and seasonal temperature fluctuations are largely influenced by elevation but moderated by a combination of landscape and physiographic features. Average diurnal variations across the sites in all seasons were influenced by elevation (DEM) with lower elevations showing larger diurnal variation (Fig 7). Furthermore, in our study site, we find that the more exposed a site is, like the ridgetops, the fluctuations are more coupled to free-air temperature than the valley bottom sites that tend to be less coupled ([9], Fig B.2 in S1 Appendix). Diurnal fluctuations in different seasons were strongly influenced by topographic position and moderated by canopy, similar to findings in previous studies [6,24,25,53].

Winter minimum temperatures were not influenced by finer physiographic features (Fig 6A). However, ridgetops showed higher summer and winter mean temperatures than the valley bottoms. Our finding is consistent with past works that suggest valley bottoms receive less radiation load in comparison to ridgetops [48,53]. Winter temperatures were found to be increasing with elevation (Fig 6). The northwest portion of the Pepperwood Preserve exhibited strong inversion patterns, and potentially is part of a large-scale cold-air pool linked to a larger valley to the northwest of our study area (Fig A.4 in S1 Appendix).

In winter and autumn, we see that maximum temperatures are driven by northness (Fig 5), which agrees with past studies on the importance of aspect for radiation load. In winter,

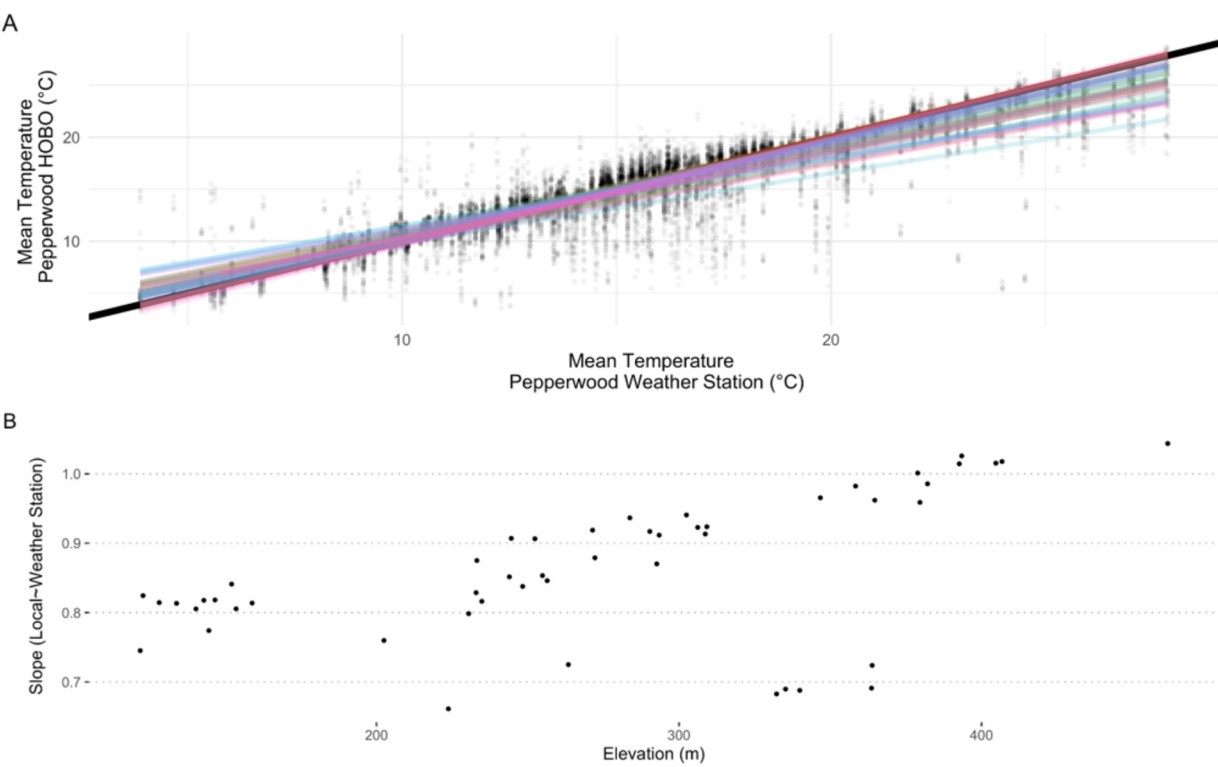

**Fig 8.** Linear fit of mean daily temperatures of HOBOs against the Pepperwood Preserve weather station (A) and slope of the relationship with elevation (B). Relationship highlighting that most of the sites are cooler in comparison to the weather station, but also show some of the high elevation sites are fundamentally aligned with the weather station climatically. Note that in panel A, dots refer to the day of the year, solid black line is 1:1, and the colored lines are individual linear fits in relation to the weather station.

maximum temperatures were only explained by northness, reflecting effects of low daytime sun angles on the difference in solar insolation between north and south-facing slopes. During the winter months, when the sun's angle is lower, there is reduced direct sunlight compared to the higher sun angles experienced in summer. This difference in sunlight exposure is likely a key factor explaining the canopy effects observed exclusively during the summer season. These effects are pronounced due to variations in canopy cover, with summer showing a greater seasonality because of the trees providing more effective shading during this period [54,55]. A recent study at Pepperwood Preserve reported that site aspect was the most important determinant of species distributions of trees [42]. But as our temperatures are from understories, mediated by canopy cover, it is most relevant to seedlings/understory vegetation rather than the overstory [56]. Our findings suggest possible stronger effects of winter minima (because of cold-air pools) that may influence future species distributions, more than summer maxima [57].

We infer that the lower elevation valley sites that also happen to be in cold-air pools would be more unstable microclimatically (larger daily fluctuations) than the high elevation sites. In our study area, low elevation valley bottom sites are strongly buffered and high elevation sites are the most coupled (used weather station as a proxy to free-air; Fig B.2 in S1 Appendix, Figs 7 and 8). A variation that is indicative of the underlying topographic heterogeneity [9,17,29,58]. Physiographic features influence the degree of coupling of a site to the free-air temperature [6,9,59].

Canopy cover in conjunction with topographic attributes plays a role in diurnal and seasonal fluctuations. We find the higher elevation sites that exhibit less diurnal and seasonal

variation have lower canopy cover as opposed to lower valley bottom sites that exhibit higher variations (diurnal and seasonal). Canopy that has been noted in many studies to be contributing to microclimatic buffering is found here to be contributing to diurnal/seasonal variability [24]. One caveat to our study is that it is possible that we did not find very strong effects of canopy cover because all our loggers were under canopies (range of 30–90%). However, as expected, the buffering effects of forest canopies are strongest when compared with an open site [22,26] (see Fig 8). So, canopy cover mediates regional and macroclimatic climatic patterns in relation with elevational gradient and is likely contributing to lower variability diurnally and seasonally [60]. Also, despite having a short canopy cover gradient, we see strong effects of canopy in the summer (Fig 5); this could be particularly important for management and conservation [61].

## Implications for global change biology

The results presented here highlight the importance of microclimates, such as valley bottoms, that may be both cooler (at night and in winter) and warmer (during the day and in summer), compared to more exposed sites that are strongly coupled with regional climate. This pattern raises two sets of questions related to impacts of future climate change. First is the question of whether sites that are more variable diurnally or seasonally (i.e., valley bottoms) will change more or less in response to regional climate change [62]. The null expectation is that warming will be equivalent across sites, so the patterns of diurnal and seasonal variability will be maintained but all sites will be warmer. In midlatitudes it is widely observed that nighttime warming is greater than daytime [63,64]. Thus, diurnal variability should be declining across the landscape, though we are not aware of direct quantification of this effect. This change may lead to species distribution shifts at cold temperature limits, such as downslope movement into valleys as cold-air pools shrink (see [65]), or (in coastal California) shifts away from the coast for species sensitive to winter frost. However, we are not aware of observations or first principles analysis addressing whether nighttime warming is greater in some sites vs. others, at a fine topographic scale.

The second set of questions is concerned with the impact of new extreme high temperatures on species [35]. The comparison of hilltop to valley bottom sites is analogous to studies of maritime vs. continental climates [66], or tropical vs. temperate climates [67], in terms of the contrasts in diurnal and seasonal means and consequences for responses to climate change. By analogy there are two new contrasting hypotheses. The first is that the buffered sites with cooler daytime and summer high temperatures will remain cooler in the future, so they will be buffered from the most extreme impacts of climate change. Alternatively, the species occupying these sites may be adapted to the narrower range of temperatures, and thus be more sensitive to rising temperatures. By contrast, those that occupy sites with high diurnal and seasonal variability may also be better adapted to withstand new extremes under a changing climate [36].

## Limitations

We want to emphasize that the study was carried out on only one year of data, so one must be cautious on drawing broad scale inferences. Secondly, the study was carried out in a particular coastal landscape, so the inferences arrived here might not be the same if a similar study is carried out at a different landscape. However, our results confirm that elevation explains most of the seasonal variation in temperature. Next, we found the influence of cold-air pools in the observed temperatures in our study system, and one needs to be aware that such phenomena affect seasonal and region-specific lapse rates, and that might not be the case in a different

study system [68]. Another point of caution relates to the potential uncertainties in the collected data due to the stratification of temperature loggers as the readings may be influenced by environment around the loggers, such as shading from canopy or being in open [40,69,70]. Additionally, the use of radiation shields can introduce bias in temperature readings, emphasizing the importance of being mindful of the type of radiation shielding employed [71]. Nonetheless, this study provides ecologists and global change biologists a way to interpret downscaled climate variables and the topographic, canopy, and seasonal interactions they may be incorporating.

## Conclusion

Our study quantifies the importance of fine physiographic features like topographic position, northness and canopy on diurnal and seasonal variability in temperatures in a topographically heterogeneous landscape in central California. With a range of about 300 m in elevation, we found that temperature inversions dominate, with warmer temperatures at higher elevations. We suggest that interactions with finer physiographic features would be key to understanding current and future species distributions. It is likely that sites having a larger diurnal or seasonal variation (valley sites in our study) might also be the same sites that are buffered from regional climatic patterns as they would be experiencing higher maximum temperatures, and thus would most likely be better adapted to withstand new extremes under a changing climate. Though this study was limited to one-year, future work spanning multiple years can be done to ascertain buffering from climate change offered by temperature refugia (different rate of warming). Buffered areas can protect native species and ecosystems from the negative effects of climate change in the short term and provide longer-term havens from climate impacts for biodiversity and ecosystem function [72,73].

## Supporting information

**S1 Appendix. Appendices. Appendix A. Fig A.1**. Fig A.1: Regression of climate variables against PC1 of physiographic space. Dominant variable explaining PC1 is DEM (elevation). **Fig A.2.** Fig A.2: Regression of climate variables against PC2 of physiographic space. Dominant variables explaining PC2 are canopy, PLP500 and northness. **Fig A.3**. Fig A.3: Custom radiation shield. **Fig A.4**. Fig A.4: Cold-air pooling evident on the northwest corner Pepperwood Preserve. (a) Mean minimum temperatures highlight inversion. (b) Season fluctuation (summer mean–winter mean) pronounced in the upper northwest corner of the Pepperwood Preserve, possibly part of larger cold-air pooling phenomena. **Table. A.1.** Table A.1 - RDA results showing climate space (only seasonal metrics) when constrained against physiographic space. Variables in bold are statistically significant (p < 0.05). **Table. A.2**. Table A.2 - Physiographic predictor variables and their effects. **Appendix B. Fig B.1.** Fig B.1: Mean temperature comparisons between HOBOs and an open site (Pepperwood Weather Station). **Fig B.2.** Fig B.2: Mean diurnal fluctuation comparisons between HOBOs and an open site (Pepperwood Weather Station) colored by elevation. **Fig B.3.** Fig B.3: Average annual temperature of each HOBO against the elevation ($R^2$ = 0.23, p-value < 0.001). **Fig B.4.** Fig B.4: Pepperwood Preserve with 50 study sites numbered.
(ZIP)

## Acknowledgments

The authors thank stewards of Pepperwood Preserve (PP) that aided in the establishment and monitoring of the plots. The authors also thank the Ackerly Lab for logistical support and

fruitful discussions. Comments from two anonymous reviewers greatly improved the final manuscript.

## Author Contributions

**Conceptualization:** Aji John, David D. Ackerly.

**Data curation:** Aji John.

**Formal analysis:** Aji John.

**Investigation:** Aji John.

**Methodology:** Aji John, Julian D. Olden, Meagan F. Oldfather, Matthew M. Kling, David D. Ackerly.

**Resources:** Aji John, David D. Ackerly.

**Software:** Aji John.

**Validation:** Aji John.

**Visualization:** Aji John.

**Writing – original draft:** Aji John.

**Writing – review & editing:** Aji John, Julian D. Olden, Meagan F. Oldfather, Matthew M. Kling, David D. Ackerly.

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
