## [Decision Letter · Decision Letter 0]

5 Nov 2023

PONE-D-23-30976Topography influences diurnal and seasonal microclimate fluctuations in hilly terrain environments of coastal CaliforniaPLOS ONE

Dear Dr. John,

Thank you for submitting your manuscript to PLOS ONE. After careful consideration, we feel that it has merit but does not fully meet PLOS ONE’s publication criteria as it currently stands. Therefore, we invite you to submit a revised version of the manuscript that addresses the points raised during the review process.

The three reviewers were supportive of this manuscript and generally in agreement about the suggested changes. In general, they would like to see some additions to the introduction section and literature review, more details on datasets, and modifications to the interpretation of results. Other reviewer comments are included in the attachments and decision.==============================

We look forward to receiving your revised manuscript.

Kind regards,

Kristofer Lasko, PhD

Academic Editor

PLOS ONE

7. We note that Figure 2 in your submission contain [map/satellite] images which may be copyrighted. All PLOS content is published under the Creative Commons Attribution License (CC BY 4.0), which means that the manuscript, images, and Supporting Information files will be freely available online, and any third party is permitted to access, download, copy, distribute, and use these materials in any way, even commercially, with proper attribution. For these reasons, we cannot publish previously copyrighted maps or satellite images created using proprietary data, such as Google software (Google Maps, Street View, and Earth). For more information, see our copyright guidelines: http://journals.plos.org/plosone/s/licenses-and-copyright.

8. We note that Figure(s)  A.3 and A.4  in your submission contain copyrighted images. All PLOS content is published under the Creative Commons Attribution License (CC BY 4.0), which means that the manuscript, images, and Supporting Information files will be freely available online, and any third party is permitted to access, download, copy, distribute, and use these materials in any way, even commercially, with proper attribution. For more information, see our copyright guidelines: http://journals.plos.org/plosone/s/licenses-and-copyright.

a. You may seek permission from the original copyright holder of Figure(s)  A.3 and A.4 to publish the content specifically under the CC BY 4.0 license. 

Reviewers' comments:

Reviewer's Responses to Questions

**Comments to the Author**

1. Is the manuscript technically sound, and do the data support the conclusions?

Reviewer #1: Yes

Reviewer #2: Yes

Reviewer #3: Yes

2. Has the statistical analysis been performed appropriately and rigorously? 

Reviewer #1: Yes

Reviewer #2: Yes

Reviewer #3: Yes

3. Have the authors made all data underlying the findings in their manuscript fully available?

Reviewer #1: Yes

Reviewer #2: Yes

Reviewer #3: Yes

4. Is the manuscript presented in an intelligible fashion and written in standard English?

Reviewer #1: Yes

Reviewer #2: Yes

Reviewer #3: Yes

5. Review Comments to the Author

Reviewer #1: The article investigates the influence of topography and canopy cover on microclimate fluctuations in hilly terrain environments of coastal California using multiple linear regression and multivariate techniques. For the 50 sites representing the diverse topography and canopy cover, the role of 18 climate variables (related to temperature), 5 topographic and canopy related variables on the diurnal fluctuation and seasonal extreme temperature is studied. Though only a year of data is used to conclude the influence of the dominant variables on seasonal fluctuations, it is clearly written in the limitation of the article. Therefore, the paper is informative and contains findings that could warrant its publication.

I have some minor suggestions in the manuscript.

1. The recent literatures related to the work needs to be included in the introduction section.

2. Figure 2: The study site numbering must be shown in the Figure 2.

3. Table 1 and Table 2: the spacing is missing between the values in table. Ex: 11.56,19.75 (15.96) space missing

4. L188: Two sites (Site 1348 and 1350) were missing – Though the site numbering is mentioned its location is details are not shown anywhere.

5. Table 3 limit the significant digits in the partial coefficients value of different climatic variables to 2 or 3.

6. Redundancy analysis in short RDA is already mentioned in line no 204, no need to mention again in line no 292.

7. L296-298, “In autumn, topographic features (DEM, PLP500, northness) and canopy are found to be statistically significant (p-value < 0.005), but in spring, topographic features DEM and northness are significant (p-value < 0.005)”. The sentence is not clear. Also, the significant variable in spring is DEM and PLP500 as per the Table A1 not the northness.

8. L321-326 , L328-331: The significance level (p-value < 0.001) is shown for every R2 value. As all the values area significant at same significance level, it can be written only once at the end.

Ex: Autumn (R2 = 65%), Spring (R2 = 28%), Summer (R2 = 41%), and Winter (R2 = 58) all significant at p-value < 0.001.

Reviewer #2: Based on the annual understory temperature changes of 50 surveyed locations, this study investigated the effects of terrain and tree cover on near surface temperature changes. The research results may help to improve our ecological understanding of fine-grained seasonal climate change in coastal environments. However, there are some concerns that the authors should address before it can be considered for publication.

1. In the last paragraph of the introduction, I suggest the authors further highlight the significance of this study.

2. In the data, the authors should add more information about data, such as DEM data availability and access.

3. I suggest the authors add more descriptions of the climate in the study area, such as multi-year average annual temperature.

4. Lines 188-190, the authors mentioned that meteorological stations 1348 and 1350 have missing values. How did the author handle the missing values?

5. Lines 213-216, the authors analyzed the average temperature of 55 stations in the study area, but this cannot reflect the impact of altitude on temperature. I suggest the authors partition the site based on altitude and compare the temperatures of different altitude sites.

6. More mechanistic explanations should be added to further explain the relationship between altitude and temperature changes.

7. A paragraph of limitation discussion should be added to clarify the limitation or uncertainty of data and methods in this current study. For example, uneven distribution of meteorological stations (Shen et al., 2014, 2018; Wang et al., 2016) may affect the research results.

References:

Spatiotemporal change of diurnal temperature range and its relationship with sunshine duration and precipitation in China. Journal of Geophysical Research: Atmospheres, 2014, 119: 13163-13179.

Weak cooling of cold extremes versus continued warming of hot extremes in China during the recent global surface warming hiatus. Journal of Geophysical Research: Atmospheres, 2018, 123: 4073-4087.

Wang W, Lu H, Yang D, et al. Modelling hydrologic processes in the Mekong River Basin using a distributed model driven by satellite precipitation and rain gauge observations. PloS one, 2016, 11(3): e0152229.

Reviewer #3: Dear authors, thanks for an interesting manuscript that deals with an important topic. The publication criteria are IMO fulfilled, but there are some clarifications needed throughout the text. Please see the attached pdf for my detailed comments.

6. PLOS authors have the option to publish the peer review history of their article (what does this mean?). If published, this will include your full peer review and any attached files.

Reviewer #1: No

Reviewer #2: No

Reviewer #3: No

---

## [Author Response · Author response to Decision Letter 0]

2 Feb 2024

We have added it in the Response to Reviewers document.

---

## [Decision Letter · Decision Letter 1]

27 Feb 2024

Topography influences diurnal and seasonal microclimate fluctuations in hilly terrain environments of coastal California

PONE-D-23-30976R1

Dear Dr. John,

We’re pleased to inform you that your manuscript has been judged scientifically suitable for publication and will be formally accepted for publication once it meets all outstanding technical requirements.

Kind regards,

Kristofer Lasko, PhD

Academic Editor

PLOS ONE

Additional Editor Comments (optional):

Reviewers' comments:

Reviewer's Responses to Questions

**Comments to the Author**

1. If the authors have adequately addressed your comments raised in a previous round of review and you feel that this manuscript is now acceptable for publication, you may indicate that here to bypass the “Comments to the Author” section, enter your conflict of interest statement in the “Confidential to Editor” section, and submit your "Accept" recommendation.

Reviewer #2: All comments have been addressed

Reviewer #3: All comments have been addressed

2. Is the manuscript technically sound, and do the data support the conclusions?

Reviewer #2: Yes

Reviewer #3: Yes

3. Has the statistical analysis been performed appropriately and rigorously? 

Reviewer #2: Yes

Reviewer #3: Yes

4. Have the authors made all data underlying the findings in their manuscript fully available?

Reviewer #2: Yes

Reviewer #3: Yes

5. Is the manuscript presented in an intelligible fashion and written in standard English?

Reviewer #2: Yes

Reviewer #3: Yes

6. Review Comments to the Author

Reviewer #2: The authors have addressed the concerns. I suggest accept this paper "Topography influences diurnal and seasonal microclimate fluctuations in hilly terrain environments of coastal California" in its present form.

Reviewer #3: (No Response)

7. PLOS authors have the option to publish the peer review history of their article (what does this mean?). If published, this will include your full peer review and any attached files.

Reviewer #2: No

Reviewer #3: No

---

## [Editor Report · Acceptance letter]

19 Mar 2024

PONE-D-23-30976R1 

PLOS ONE

Dear Dr. John, 

I'm pleased to inform you that your manuscript has been deemed suitable for publication in PLOS ONE. Congratulations! Your manuscript is now being handed over to our production team.

Kind regards, 

on behalf of

Dr. Kristofer Lasko 

Academic Editor

PLOS ONE